# A Hypothesis and Evidence That Mercury May be an Etiological Factor in Alzheimer’s Disease

**DOI:** 10.3390/ijerph16245152

**Published:** 2019-12-17

**Authors:** Robert Siblerud, Joachim Mutter, Elaine Moore, Johannes Naumann, Harald Walach

**Affiliations:** 1Rocky Mountain Research Institute, 9435 Olsen Court, Wellington, CO 80549, USA; 2Environmental Medicine, 78467 Konstanze, Germany; Jo.mutter@web.de; 3Memorial Hospital, Colorado Springs, CO 80549 (Retired), USA; Mooredaisyelaine@aol.com; 4European Institute for Physical Therapy and Balneology, Stadtsr 7, D-79104 Freiburg, Germany; dr.j.naumann@gmail.com; 5Department of Psychology, University Witten-Herdecke, 58455 Witten, Germany; harald.walach@uni-wh.de

**Keywords:** Alzheimer’s disease, mercury, etiology

## Abstract

Mercury is one of the most toxic elements and causes a multitude of health problems. It is ten times more toxic to neurons than lead. This study was created to determine if mercury could be causing Alzheimer’s disease (AD) by cross referencing the effects of mercury with 70 factors associated with AD. The results found that all these factors could be attributed to mercury. The hallmark changes in AD include plaques, beta amyloid protein, neurofibrillary tangles, phosphorylated tau protein, and memory loss—all changes that can be caused by mercury. Neurotransmitters such as acetylcholine, serotonin, dopamine, glutamate, and norepinephrine are inhibited in patients with Alzheimer’s disease, with the same inhibition occurring in mercury toxicity. Enzyme dysfunction in patients with Alzheimer’s disease include BACE 1, gamma secretase, cyclooxygenase-2, cytochrome-c-oxidase, protein kinases, monoamine oxidase, nitric oxide synthetase, acetyl choline transferase, and caspases, all which can be explained by mercury toxicity. Immune and inflammatory responses seen in patients with Alzheimer’s disease also occur when cells are exposed to mercury, including complement activation, cytokine expression, production of glial fibrillary acid protein antibodies and interleukin-1, transforming growth factor, beta 2 microglobulins, and phosphodiesterase 4 stimulation. Genetic factors in patients with Alzheimer’s disease are also associated with mercury. Apolipoprotein E 4 allele increases the toxicity of mercury. Mercury can inhibit DNA synthesis in the hippocampus, and has been associated with genetic mutations of presenilin 1 and 2, found in AD. The abnormalities of minerals and vitamins, specifically aluminum, calcium, copper, iron, magnesium, selenium, zinc, and vitamins B1, B12, E, and C, that occur in patients with Alzheimer’s disease, also occur in mercury toxicity. Aluminum has been found to increase mercury’s toxicity. Likewise, similar biochemical factors in AD are affected by mercury, including changes in blood levels of homocysteine, arachidonic acid, DHEA sulfate, glutathione, hydrogen peroxide, glycosamine glycans, acetyl-L carnitine, melatonin, and HDL. Other factors seen in Alzheimer’s disease, such as increased platelet activation, poor odor identification, hypertension, depression, increased incidences of herpes virus and chlamydia infections, also occur in mercury exposure. In addition, patients diagnosed with Alzheimer’s disease exhibit higher levels of brain mercury, blood mercury, and tissue mercury in some studies. The greatest exogenous sources of brain mercury come from dental amalgams. Conclusion: This review of the literature strongly suggests that mercury can be a cause of Alzheimer’s Disease.

## 1. Introduction

The cause of Alzheimer’s disease [AD] has been one of the great medical mysteries since Alois Alzheimer first identified this disease in 1906. It is the fourth leading cause of death in the United States, affecting 4.2 million citizens. Although recent studies have shown a decline in incidence, it is still one of the most unmet challenges of our age [1]. The worldwide dementia prevalence is over 44 million people, with an annual cost of 604 billion USD. It is estimated that the worldwide prevalence will triple to 135.5 million by 2050. At least 30% to 50% of all individuals above the age of 85 are affected by AD in the industrialized countries [2]. Approximately three to five percent of all cases have a genetic factor [3]. Many researchers believe that some environmental factor is responsible for the etiology of Alzheimer’s disease.

The amount of neurofibrillary tangles found in affected Alzheimer brain regions correlate with the severity of the disease [3]. Some studies suggest that fibrillary nerve cell changes may begin to occur as early as 50 years before the onset of clinical symptoms. This would rule out old age as the cause. Neurofibrillary tangles in low amounts are found in about 20% of the population aged 20 to 30 years without clinical symptoms. By 70 to 80 years, 90% of individuals display neurofibrillary tangles in the brain [3]. Of this age group, the 35% who have the highest number of histological detectable neurofibrillary tangles suffer clinically from symptoms of Alzheimer’s disease.

Over the past three decades, a number of studies have suggested a pathological role of inorganic mercury as being one of the causes of Alzheimer’s disease [4,5,6]. This is underlined by the fact that mercury is ten times more toxic to neurons than lead, another neurotoxin [7]. Only mercury, in low levels, was able to induce the hallmark changes seen in AD, compared to cadmium, manganese, aluminum, iron, and lead [6]. A substantial number of neurons, interneurons, and CNS-endothelial cells derived from people who die from a neuro-degenerative disease are full of intracellular mercury deposits, demonstrating that mercury easily penetrates the blood brain barrier, and enters neurons, resulting in neuronal loss [8].

Mercury exposure has risen worldwide in recent decades due to industrial activities, gold mining, medicinal usages, and the burning of fossil fuels. Mercury released in the biosphere accumulates and is additive over thousands of years, and the content in the biosphere is today two- to five-fold that of the pre-industrial era [9]. Mercury levels accumulate in organisms at the end of the food chain, such as predatory fish and humans. The mercury content in tuna is rising four percent per year [10] and has risen more than 100- to 1000-fold over the past 1000 years. Besides fish consumption, an important anthropogenic source of mercury originates from dental amalgams, which are comprised of about 50 percent mercury. The World Health Organization (WHO) stated in 1991 that the largest average daily intake of mercury in the general population originates from the dental amalgam [11]. Other medicinal sources are mercury-based preservatives used in medications such as vaccines [12]. The World Health Organization has stated “there is no safe level of mercury.” The purpose of this paper is not to identify the sources of mercury, but to examine all known physiological changes that occur in Alzheimer’s disease and determine if mercury could be causing these changes.

## 2. Methodology

To identify what physiological and pathological changes occur in Alzheimer’s disease, a comprehensive AD book entitled *Encyclopedia of Alzheimer’s Disease* [13], researched and written by Elaine Moore, was used as the primary reference for a summary of the pathological and physiological changes that occur in Alzheimer’s disease. Each of these AD changes was identified and then cross-referenced with mercury databases and PubMed to identify scientific articles that would explain whether, and, if so, how, mercury could cause these changes. Positive and negative reports were considered equally, and all data presented here helped explore the hypothesis. To our surprise, all 70 factors identified to date as occurring in Alzheimer’s disease can be explained by mercury toxicity. The following is a brief summary of the physiological and pathological changes occurring in AD, followed by evidence describing how mercury could cause them.

## 3. Results

### 3.1. A. Hallmark Brain Changes in Alzheimer’s Disease (AD)

Three of the major diagnostic markers of AD, including neurofibrillary tangles, secretion of beta amyloid protein, and hyperphosphorylation of tau protein, can all be explained by mercury toxicity.

**Senile Plaques:** Neuritic plaques are composed of neuritic protein deposits found in excess amounts in AD brains. They are composed of beta amyloid protein [13].***Mercury*:** Mercury stimulates the formation of beta-amyloid protein, which plays a role in the pathogenesis of AD by causing oxidative stress and neurotoxicity [14,15,16].**Amyloid Precursor Protein (APP):** APP is the parent protein from which beta amyloid is derived. APP is broken down into fragments. A defect in APP may cause AD [13].***Mercury*:** Inorganic mercury reduces the level of APP [17], and strongly inhibits gamma secretase processing of APP [18].**Tau Protein:** Accumulated phosphorylated tau protein is responsible for neurodegeneration in AD (13).***Mercury:*** Mercury can significantly increase the phosphorylation of tau protein [19,20].**Neurofibrillary Tangles (NFT):** NFTs are a characteristic brain finding in AD, consisting of a hyperphosphorylated tau protein [13].***Mercury:*** Mercury increases the phosphorylation of tau protein, resulting in NFTs [19].**Memory Loss:** Loss of memory is the hallmark symptom of AD [13].***Mercury:*** Memory loss is a hallmark symptom of mercury toxicity [21].

### 3.2. B. Neurotransmitters

Most major neurotransmitter functions are disturbed in AD. Likewise, in mercury toxicity, the same neurotransmitters are affected.

**Acetylcholine:** Levels of acetylcholine are reduced in AD [13].**Mercury:** Acetylcholine is reduced in mercury toxicity [22].**Serotonin:** AD brains have a serotonin deficit [13].**Mercury:** Mercury inhibits the binding of serotonin to brain receptors [23].**Dopamine:** Dopamine levels in the AD brain are low, due to a deficiency of dopamine D2 receptors [13].**Mercury:** A negative correlation was found between total mercury and dopamine D2 receptors in the brain of wild mink [24].**Glutamate:** Excess glutamate leads to neurodegeneration in AD [13].**Mercury:** Mercury inhibits glutamate uptake and stimulates release of glutamate [25].**Nitric Oxide:** Excess nitric oxide can contribute to AD [13].**Mercury:** Mercury chloride induces nitric oxide synthetase [26].**S-Adenosylmethione (SAMe):** AD patients with depression have decreased levels of SAMe [13].**Mercury:** Mercury inhibits SAMe [27].**Norepinephrine:** AD brains have decreased levels of norepinephrine [13].**Mercury:** Mercury decreases norepinephrine in brain synapses [28].

### 3.3. C. Enzymes

Mercury’s ability to interact with enzymes is due to its affinity to attach to sulfhydryl groups, which helps to explain its toxicity.

**BACE 1 (Beta Amyloid Cleaving Enzyme):** BACE 1 has been implicated as the enzyme responsible for plaque in AD brains [13].**Mercury:** Methyl mercury increases the end products of BACE 1 [17,29].**Gamma Secretase:** Early-onset AD patients have mutations in genes that produce gamma secretase, which increases beta amyloid protein [13].**Mercury:** Mercury inhibits gamma secretase, suggesting the inhibition contributes to mercury-induced neuron toxicity [30].**Kinases:** Kinases are protein enzymes responsible for phosphorylation of hydroxyl proteins. The enzyme has a high concentration in AD [13].**Mercury:** Mercury induces hyperphosphorylation by activating kinase pathways [20].**Cyclooxygenase-2 (Cox-2):** Cox-2 is released during inflammation and is over-expressed in AD [13].**Mercury:** Mercury induces expression of cox-2 [31].**Cytochrome-c-oxidase:** Deficiencies of cytochrome-c-oxidase have been reported in AD patients [13].**Mercury:** Mercuric chloride inhibits cytochrome-c-oxidase [32].**Monaamine Oxidase (MAO):** MAO forms hydrogen peroxide which is higher in AD [13].**Mercury:** Mercury increases MAO activity [33].**Nitric Oxide Synthetase:** Elevated levels of nitric oxide synthetase contribute to AD [13].**Mercury:** Mercury induces nitric oxide synthetase [34].**Acetyl Choline Transferase (CAT):** CAT is reduced by up to 90 percent in AD [13].**Mercury:** Mercury inhibits CAT [35].**Caspases:** In AD, there is activation of apoptosis caspases that underlie the pathology of AD [13].**Mercury:** Mercury increases activated caspases [36].

### 3.4. D. Immunity

Alzheimer’s disease is also known to be caused or aggravated by over-activation of the immune system and inflammation.

**Complement:** AD is triggered by the activation of complement 1 [13].**Mercury:** Studies show a correlation between mercury vapor and complement [37].**Cytokines:** Cytokines contribute to inflammation. The beta amyloid in AD inflames the surrounding microglial cells [13].**Mercury:** Mercury significantly induces cytokine expression [38].**Glial Fibrillary Acid Protein Antibodies (GFAP):** GFAP are autoantibodies associated with AD [13].**Mercury:** Methyl mercury induces GFAP [39].**Interleukin 1 (IL-1):** Over-expression of IL-1 in AD sets the cytokine cycle in motion [13].**Mercury:** Mercuric chloride increases the release of IL-1 [40].**Transforming Growth Factor Beta 1 (TGF B-1):** TGF B-1 regulates beta amyloid precursor protein synthesis. It is a cytokine that plays a central role in AD [13].**Mercury:** Thimerosal contains ethyl mercury, which enhances the expression of TGF B-1 [41].**Tumor Necrosis Factor (TNF):** Is a cytokine that contributes to AD. Excess levels of TNF are found in the cerebral fluid of AD [13].**Mercury:** Low levels of mercuric chloride increase the release of TNF [42].**Beta-2 Microglobulin (B-2 M):** B-2 Microglobulin is a peptide that is increased in the amyloid disorder of AD [13].**Mercury:** B-2 Microglobulins are used as an early indicator of mercury toxicity in the kidney [43].**Inflammation:** AD brains show evidence of inflammation adjacent to plaques [13].**Mercury:** Mercury exposure increases pro-inflammatory cytokines [42].**Phosphodiesterase 4 (PDE 4):** PDE 4 is an enzyme that degrades cyclic amp (CAMP). CAMP underlies memory formation [13].**Mercury:** Mercury has been found to stimulate PDE 4 and reduce CAMP [44].

### 3.5. E. Genetic

Studies suggest that AD has both a genetic component and environmental etiology.

**Apolipoprotein E4 (APOE4)**: APOE4 may account for 60% of all AD cases. APOE 4 appears to promote the binding of the amyloid beta protein, facilitating the formation of plaque [13].**Mercury:** APOE4 has a reduced ability to bind mercury, and thus potentiates mercury damage [3].**Genetic Mutations:** Nearly 30% of early-onset AD is linked to the presenilin 1 gene [13].**Mercury:** Mercury is known to cause gene mutations [45].**Presenilin Gene Mutation:** In the Antioguia region of Columbia, families with the Paisa presenilin 1 gene mutation develop AD nearly ten years earlier than individuals with the exact same mutation in Japan [46].**Mercury:** In the Antioguia region, individuals are exposed to the highest levels of mercury in the world from gold mines [47].**Alpha-2 Macroglobulins (A2M):** A2M is a gene suspected of controlling the rate of beta amyloid protein production and is a possible susceptible gene for AD [13].**Mercury:** A2M levels were found to be significantly higher in mercury exposed workers [48].**Hippocampus DNA Synthesis:** AD is characterized by the death of cells in the hippocampus [13].**Mercury:** Mercury inhibits DNA synthesis by 44 percent in the hippocampus of rats [49].

### 3.6. F. Minerals

**Aluminum:** Studies suggest an association between AD and aluminum [13].**Mercury:** Aluminum dramatically enhances mercury toxicity [50].**Calcium:** Amyloid beta protein is increased in AD, which disturbs calcium concentrations and the calcium metabolism [13].**Mercury:** Mercury increases intracellular calcium concentrations [51].**Copper:** The Nun Study on AD found an association between high serum copper and AD [13].**Mercury:** Mercury can tie up metallothionein so the body cannot clear out toxic metals such as copper [52].**Iron:** Increased levels of the iron-binding protein melanotransferrin is seen in AD. Iron deposits are found in senile plaques. Beta amyloid production is increased in the presence of iron [13].**Mercury:** Iron levels in the blood have been correlated with mercury and memory [53].**Magnesium:** Magnesium levels are depleted within the hippocampus of AD [13].**Mercury:** Mercury competes with magnesium and interferes with magnesium-dependent pathways [53].**Zinc:** Zinc tends to accumulate in the areas of the brain most prone to AD damage. Zinc is suspected of aggregating beta amyloid deposits and pulls copper into the deposits. Zinc levels in the hippocampus are decreased in AD [13].**Mercury:** Mercury tends to displace zinc in energy producing pathways for neurotransmitters because they have a similar chemical structure [54].**Selenium:** Studies have shown that AD patients have lower levels of selenium in their plasma [55].**Mercury:** Selenium binds to mercury and neutralizes toxicity. Workers in mercury mines have lower levels of selenium in their blood [56].

### 3.7. H. Vitamins

**Folic Acid/Folate:** Low levels of folic acid increase the risk of AD [13].**Mercury:** Studies have found a negative correlation between serum folate and blood mercury [57];**Thiamine (Vitamin B1):** Low levels of vitamin B1 increase the risk of developing AD [13].**Mercury:** Mercury reduces brain levels of vitamin B1 [58].**Vitamin B12:** AD is characterized by vitamin B12 deficiencies [13].**Mercury:** Mercury can reduce the uptake of vitamin B12 [59].**Vitamin C:** Evidence suggests that vitamin C may prevent the onset of AD [13].**Mercury:** Workers exposed to mercury had lower levels of plasma ascorbic acid [60].**Vitamin E:** Vitamin E deficiencies have been associated with AD [13].**Mercury:** Vitamin E provides complete protection from mercury toxicity in poisoned rats [61].**Vitamin D:** Vitamin D may play a role in the pathophysiology of cognition decline in AD [13].**Mercury:** Little research has been done with mercury and vitamin D. Vitamin D deficiencies in AD may be explained by a lack of sun exposure. Vitamin D is anti-inflammatory, and mercury is known to cause inflammation.

### 3.8. I Amino Acids, Antioxidants, and Biochemicals

**Homocysteine:** Homocysteine is elevated in AD and is toxic to the brain. It is associated with brain shrinkage [13].**Mercury:** Mercury toxicity overwhelms SAMe, resulting in elevated homocysteine [62].**Arachidonic Acid:** Arachidonic acid induces the polymerization of tau and induces apoptosis in neurons [13].**Mercury:** Methyl mercury induces the production of arachidonic acid in cerebellar granule cells [63].**DHEA Sulfate:** The ratio of DHEA sulfate to cortisol is significantly lower in AD [13].**Mercury:** Mercury contributes to lower levels of DHEA [64].**Glutathione:** Glutathione is essential for removing free radicals from the brain [13].**Mercury:** Rat studies have shown that mercury inhibits glutathione. In mercury toxicity, levels of glutathione decline and brain damage occurs [65].**Antioxidants:** Free radicals are responsible for oxidative cell death associated with the amyloid beta protein in AD [13].**Mercury:** Mercury produces free oxygen radicals that cause nerve damage [65].**Hydrogen Peroxide:** Beta amyloid protein in AD increases the production of hydrogen peroxide [13].**Mercury:** The uptake of mercury vapor into cells is increased by hydrogen peroxide [66].**Lipid Peroxidation:** The accumulation of lipid peroxidation products has been found in AD brains [13].**Mercury:** Mercury promotes lipid peroxidation [67].**Glycosaminoglycans (GAGs):** GAGs are found in AD brain plaques. Sulfated GAGs have an affinity for beta amyloid [13].**Mercury:** Rat studies have shown that mercuric chloride increases serum GAGs [68].**Brain Derived Neurotrophic Factor (BDNF):** BDNF is a cytokine that influences nerve growth in brains [13].**Mercury:** Rat studies have shown methyl mercury-induced cell death occurs by way of receptors of BDNF [69].**Melatonin:** Melatonin levels, which regulate sleep and wakefulness, are low in AD [13].**Mercury:** Melatonin protects against mercury toxicity [70].**High Density Lipoprotein (HDL):** HDL, the good cholesterol, reduces the risk of developing AD [13].**Mercury:** Low HDL cholesterol is associated with high blood mercury [71].

### 3.9. J. Other Factors

**Platelets:** In AD, platelet activation rates are 30% to 50% higher compared to non-AD subjects [13].**Mercury:** Normal platelet function is affected by high and low concentrations of mercury. Low mercury levels encourage clotting of red blood cells [72].**Odor Identification:** Subjects who have a mild cognition impairment and poor odor identification scores are more likely to develop AD [13].**Mercury:** Loss of smell is a symptom of mercury toxicity [73].**Smoking:** Smoking increases the risk of AD [13].**Mercury:** Dopamine levels are reduced in AD. Smoking increases neurotransmitters, such as dopamine, that can be seen as self-medicating [74]. One study found that subjects with mercury dental amalgams smoked significantly more than a control group without amalgams who were age and sex matched. Animal studies have shown a negative correlation between total mercury and dopamine D2 receptors (24).**Herpes Virus:** A high percentage of AD brains contain latent herpes simplex virus [13].**Mercury:** Mercury exposure increases herpes virus replication [75].**Chlamydia Pneumonia:** Chlamydia bacterium has been found in parts of the brain of late-onset AD [13].**Mercury:** Mercury has been found to increase chlamydia infections in animals [76].**African Americans:** AD is very rare in west Africa. In the U.S., African Americans have an AD rate two times higher than whites [13].**Mercury:** In lakes across tropical Africa, fish have low levels of mercury [77].**Depression:** Depression is more likely to occur in mild to moderate AD [13].**Mercury:** A major symptom of mercury toxicity is depression [78]. Studies have found that subjects with mercury dental amalgams suffered significantly more from depression than subjects without amalgams who were age and sex matched.**Hypertension:** Chronic hypertension increases the risk of AD [13].**Mercury:** Serum mercury concentrations are associated with hypertension [79].**Parkinson’s Disease (PD):** Up to one third of PD patients develop AD [13].**Mercury:** Many epidemiological studies have shown an association between PD and exposure to mercury [80];**Down Syndrome:** All patients with Down Syndrome will develop neuropathological hallmarks of AD [13];**Mercury:** An Egyptian study found elevated serum mercury levels in the Down Syndrome group compared to a control group, and they had a significant increase in DNA damage [81].

## 4. Discussion

Mercury is a persistent, bio-accumulative, neurotoxic metal that can accumulate in the brain. Since industrialization, mercury in the air and water has increased by 3- to 5-fold, and between 1977 and 2002 mercury increased in fish 4- to 5-fold [10]. Mercury in fish increases by about four percent every year.

The half-life of mercury in the brain is from several years to decades [82]. The World Health Organization rates mercury as one of the ten most dangerous chemicals to public health, and the U.S. Agency for Toxic Substances rates it in the top third. A study on autopsied brains by Weiner, et al., found that mercury concentrations in the occipital region of the brain increased with age [83]. Mercury exposure in humans is mainly derived from fish consumption, dental amalgams, and mercury-based vaccines.

For decades, there has been a controversy regarding whether mercury exposure from these sources has clinical relevance. For fish consumption, there are confusing observations regarding healthy outcomes, because some kinds of fish are also big sources of selenium and omega 3 fatty acids, which combat mercury toxicity. Additionally, the chemical form of mercury in fish already has reacted to molecules like cysteine or selenium. It may be much less toxic than the form which is used in toxicology studies or is produced in the gastrointestinal tract of humans by the methylation of inorganic mercury from dental amalgams [84,85]. Surprisingly, mercury vapor seems to be more toxic in rats than unreacted methyl mercury [86]. The question arises, which are the main sources of mercury in the human brain? The monitoring of blood, urine, or other biological specimens in living organisms has not correlated appropriately with brain mercury content.

Pendergrass exposed rats to mercury vapor and found average rat brain mercury concentrations increased significantly. The identical neural chemical lesions in the rat brain were similar to or of greater magnitude than those seen in AD brains. It was concluded that low levels of mercury vapor can inhibit the polymerization of brain tubulin essential for the formation of microtubules [87].

Blood mercury levels in patients (N = 33) with AD were compared to a control group with major depression (N = 45) and another control group (N = 65) with non-psychiatric disorders. In early-onset AD (N = 13), blood levels were almost three-fold higher than the controls. Blood mercury levels were more than two-fold higher in the whole AD group compared to the control group [88].

Studies have found mercury causes blood–brain barrier damage [89]. Once damaged, toxins can enter the brain with greater ease. Vascular disorders have also been associated with AD, and it is known that mercury affects the cardiovascular system. Methyl mercury has been shown to damage the central nervous system and is associated with vascular dysfunction, hemorrhage, and edema of the brain [90]. Research has shown mercury exposure can produce acute and chronic hypertension [91]. One study found that subjects with mercury dental amalgams had significantly higher blood pressure, both systolic and diastolic, compared to an age and sex matched control group without dental amalgams [92].

A review of the literature in a recent study [92] has confirmed many of this study’s findings regarding the relationship of mercury with AD, suggesting mercury may be an etiological factor in AD.

The incidence of AD is on the decline (1). Could this be due to the decreased use of mercury dental amalgams? Sweden and Denmark are countries that have banned amalgams. In the United States, estimates by the International Academy of Oral Medicine and Toxicology state that approximately 50% of dentists still use dental amalgams, a great reduction from the last four decades, and many other dentists have reduced their use of them.

If mercury is causing AD, it is important to understand its pathway to the brain. According to the World Health Organization (11), the greatest source of mercury originates from the dental amalgam, which contains 50% mercury. Mercury vapor is continuously released from the amalgam in the non-ionized state [93] and then inhaled. It then enters the lungs, and 80% of the mercury is taken up by the blood. Mercury then easily crosses the blood–brain barrier and becomes ionized. Once inside the brain, it becomes trapped, and has a half-life of up to several decades. If mercury is involved, it can cause pathological changes in the brain decades before AD is diagnosed.

Evidence has been presented in this paper that explains how mercury can cause nearly all the physiological and pathological changes that occur in AD. Seventy factors associated with AD were examined, and all can be explained by mercury toxicity. (Refer to Table 1 for synopsis) Statistical analysis by the Statistics Laboratory at Colorado State University found the probability of all 70 Alzheimer changes that could be explained by mercury toxicity (70/70) had a probability (p) of happening by chance of 0.0001 or less.

Some of the studies referred to involve animals or were in-vitro experiments, and one cannot be sure if they are relevant to humans. However, the purpose of this paper was to show a relationship between mercury and pathogenic factors of AD, and this was shown to be the case.

## 5. Conclusions

The evidence suggests that mercury could be a factor in Alzheimer’s disease. Mercury exposure has risen worldwide in the last decades. This is due to the fact that mercury, which is released in the biosphere through industrial activities, medicinal usages, and the burning of fossil fuels, is not able to degrade. Thus, all mercury released into the biosphere will accumulate and become additive for thousands of years.

Typical brain damage in AD commences 20 to 50 years before symptoms manifest. Most of the U.S. population has been exposed to mercury from several sources, from prenatal to postnatal. Further research is needed to explore the hypothesis that mercury may be an etiological factor in AD.

## Figures and Tables

**Table 1 ijerph-16-05152-t001:** Synopsis: Similar changes that occur in both Alzheimer’s Disease and Mercury Toxicity.

**A. Hallmark Changes**
1. Senile Plaques—both increased;
2. Amyloid Precursor Protein—both decreased;
3. Tau Protein—both increased;
4. Neurofibrillary Tangles—both increased;
5. Memory—both decreased.
**B. Neurotransmitters**
1. Acetylcholine—both decreased;
2. Serotonin—both decreased;
3. Dopamine—both decreased;
4. Glutamate—both increased;
5. Nitric Oxide—both increased;
6. S Adenosylmethione (SAMe)—both decreased;
7. Norepinephrine—both decreased.
**C. Enzymes**
1. BACE 1 (Beta Amyloid Clearing Enzyme)—both increased;
2. Gamma Secretase—both decreased;
3. Caseine Kinase (CK-P)—both decreased;
4. Cyclooxygenase-2 (Cox-2)—both increased;
5. Cytochrome C Oxidase—both decreased;
6. Kinases—both increased;
7. Monaamine Oxidase (MAO)—both increased;
8. Nitric Oxide Synthetase (NOS)—both increased;
9. Capases—both increased.
**D. Immunity**
1. Complement—both increased;
2. Cytokines—both increased;
3. Glial Fibrillary Acid Protein Antibodies (GFAP)—both increased;
4. Interleukin 1 (IL-1)—both increased;
5. Transforming Growth Factor Beta (TGF B1)—both increased;
6. Tumor Necrosis Factor (TNF)—both increased;
7. Beta-2 Microglobulin—both increased;
8. Inflammation—both increased;
9. Phosphodiesterase 4 (PPE-4)—both decreased.
**E. Genetic**
1. Apolipoptotein E4—both related;
2. Genetic Mutations—both related;
3. Alpha 2 Macroglobulin (A2M)—both increased;
4. Hippocampus DNA Synthesis—both related.
**F. Minerals**
1. Aluminum—both related;
2. Calcium—both increased;
3. Copper—both related;
4. Iron—both related;
5. Magnesium—both decreased;
6. Zinc—both related;
7. Selenium—both decreased.
**G. Vitamins**
1. Folic acid/folate—both decreased;
2. Thiamine—both decreased;
3. Vitamin B12—both decreased;
4. Vitamin C—both decreased;
5. Vitamin E—both decreased;
6. Vitamin D—both related.
**H. Amino Acids, Antioxidants, and Biochemical Changes**
1. Homocysteine—both increased;
2. Arachidonic Acid—both increased;
3. DHEA Sulfate—both decreased;
4. Glutathione—both decreased;
5. Antioxidants—both related;
6. Hydrogen Peroxide—both decreased;
7. Lipid Peroxidation—both increased;
8. Glycoaminoglycans—both increased;
9. Acetyl-L Carnitine—both related;
10. Brain Derived Neurotrohic Factor (BDNF)—both related;
11. Melatonin—both decreased;
12. High Density Lipoprotein—both decreased.
**I. Other Factors**
1. Platelets—both increased;
2. Odor Identification—both decreased;
3. Smoking—both increased;
4. Herpes Virus—both increased;
5. Chlamydia Pneumonia—both increased;
6. African Americans—both related;
7. Depression—both increased;
8. Hypertension—both increased;
9. Parkinsons Disease—both increased;
10. Downs Syndrome—both increased.

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
