# Peer review of "A Hypothesis and Evidence That Mercury May be an Etiological Factor in Alzheimer’s Disease"

_ijerph, 2019, doi:10.3390/ijerph16245152_

Round 1

Reviewer 1 Report

In this work, the authors  explored the hypothesis that mercury may be a cause of Alzheimer’s disease by cross referencing the effects of mercury with 70 factors associated with AD. They use "Encyclopedia of Alzheimer’s Disease" as a primary source to identify  several factors of AD. The hypothesis is interesting, however a quantitative meta-analysis design is needed to support the link between AD and mercury. It is recommended to redesign the methodological analysis to assert the hypothesis with statistical significance.

Author Response

1Reviewer One

Suggestion 1: Redesign methodological analysis to assert the hypothesis with statistical significance.

Response: I asked the Statistics Lab at Colorado State University to do an analysis on the probability of all seventy changes with AD that could be explained by mercury,

The p value was 0.0001. I included that in the paper corrections.

Reviewer 2 Report

This article by Siblerud et al. summarizes all the literature linking mercury and Alzheimer’s disease (AD) and concludes that mercury may be an etiological factor contributing to AD. Whether or not environmental factors (e.g., air pollution) contribute to AD pathophysiology is a trending topic nowadays. Several funded studies are currently investigating the possible key players and biological mechanisms involved. Mercury turned out to be one of them. Below are some comments that would strengthen the current article although overall well-written and timely important.

Although the introduction, methodology, and discussion sections are well-written, the abstract and results sections are written as a ‘list’/’bullet points’ of evidence which clearly dampened my enthusiasm. The authors should think about restructuring the results section and provide a nice flow between all the evidence pointing mercury out instead of giving a surprising and not-common bullet point format.

The abstract needs to be rewritten as well to better reflect the content of the current review article instead of providing, again, the list of evidence putting mercury as the villain.

The vascular contribution to AD has been omitted in the current manuscript. There are growing evidence showing that blood-brain barrier and brain perfusion dysfunctions may happen at a very early stage, years prior the AD hallmarks (amyloid plaques and tau tangles) and cognitive decline appear. The authors should at least mention it and cite the relevant recent literature. For instance, PMIDs 25611508, 30643288, and 30642436.

The authors should be aware of a recent review article from early 2019 (Insights into the Potential Role of Mercury in Alzheimer's Disease. Bjørklund G, Tinkov AA, Dadar M, Rahman MM, Chirumbolo S, Skalny AV, Skalnaya MG, Haley BE, Ajsuvakova OP, Aaseth J. J Mol Neurosci. 2019 Apr;67(4):511-533. doi: 10.1007/s12031-019-01274-3. Epub 2019 Mar 15. Review. PMID:30877448), which discusses all the data supporting mercury being a key player in the course of AD. Citing this review and explaining how the current one is different would be useful.

Author Response

Reviewer Two:

Suggestion 1: Restructure the results section and provide a flow between all the evidence pointing mercury out.

Response: I did highlight the subheadings of the results section in bold. Hopefully it will make it more readable.

      The results section had been condensed from the original manuscript which goes into more detail. I probably reads easier but greatly adds to the length of the paper with about 75 more references. If necessary, I would be glad to submit the longer format.

Suggestion 2: The abstract needs to be rewritten.

Response: I rewrote the abstract and hopefully have made it easier to read. I am not sure how to rewrite the abstract without mentioning all the AD changes and linking them with mercury.

Suggestion 3: The vascular contribution to AD.    

Response: I added the relationship of the vascular system, blood brain barrier, and early brain changes prior to AD diagnosis and how they could be related to mercury. It was added in the discussion section

Suggestion 4: Respond to the Bjorklund paper.

Response: I did mention the Bjorklund paper, and feel my paper is more complete and provides more evidence on a probable source of AD mercury. 

Reviewer 3 Report

The manuscript entitled “A Hypothesis and Evidence That Mercury May Cause Alzheimer’s Disease” done by Robert et al., represents highlights the effect of mercury on Alzheimer’s disease the serious health concerns with regard to Hallmark changes in AD. He collected the mercury related literature and research reports from PubMed database on AD toxicity, involving gamma secretase, nitric oxide synthetase and acetylcholine transferase and Immune responses. The most important thing in this article the authors elaborately discussed the  abnormalities of minerals and vitamins, specifically Al, Ca, Cu, Fe, Mg, Se, Zn, and vitamins B1, B12, E, and C, occur in both AD and mercury toxicity.

This review will draw with good scientific conclusions. This is an interesting review focusing on Mercury toxicity how influences by triggering the Genetic factors in AD. How mercury toxic and interacting with other proteins, enzymes, neurotransmitters, and vitamins and minerals will affect during the exposure related study repots data has been successfully collected. It is well-established serious pollutants however; it is deserve a serious health concern in prospects of mercury exposed toxicity of AD patents are more concerns.

Mercury exposure-related data collection and search strategy are appreciable and data extraction and quality evaluation worthy in the scientific domain. Mercury exposure more important for assessing the health risks from AD related diseases are more concerns. The authors collected information related to mercury toxicity affected genes, enzymes, metabolism is well deserved, and it is important to protect from these harmful mercury exposure in present living lifestyle.

 Some concerns:

Write tabular for the list of genes enzymes of other related information on mercury exposure correlated with latest reports on mercury related AD. Draw some pathway related illustration influences of mercury toxicity will be more attract reader to understand.

The author can explain these minor concerns with scientific explanations will be added more value to this public health prospective.

Author Response

Reviewer Three:

Suggestion 1: Write a tabular list with AD changes and the mercury connection.

Response: I created a table giving a synopsis of AD changes and mercury.

Suggestion 2: Draw some pathway illustrating the influence on the mercury toxicity.

Response: I added the pathway of mercury into the body and how it is toxic to the body.

Round 2

Reviewer 1 Report

The authors did not clarify the request of this reviewer.

They only reported:

"Statistical analysis by the Statistics Laboratory at Colorado State University found the probability of all 70 Alzheimer changes that could be explained by mercury toxicity (70 of 70) had a probability (p) of happening by chance of 0.0001 or less."

in the new version of the manuscript without explaining how the Statistics Laboratory conducted this analysis. 

Please, add some details about the statistical analysis.

Author Response

Suggestion: Please add some details about the statistical analysis.

Response: Seventy physiological and pathological changes were identified in Alzheimer’s disease. All seventy could be explained by mercury toxicity. The statistical probability of this happening by chance was determined by the Statistics Laboratory at Colorado State university. The chance of this happening by chance was less than one in ten thousand (p=0.0001).

Reviewer 2 Report

I am still bothered with the bullet point style of the results section. If this is a requirement from the journal itself I'm okay, but otherwise, please provide constructed paragraphs instead of a list of articles.

Lastly, I'm surprised that the authors did not cite the important papers demonstrating the vascular contribution to AD. I previously provided 3 PMIDs that the authors should consider.

Otherwise, the abstract is now clearer.

Author Response

Suggestion: The reviewer was bothered by the bullet point style of the results.

Response: The Alzheimer’s disease change was described in a short paragraph. This was followed by a short sentence (or short paragraph) to determine if mercury could be associated with the particular change. In all cases there was evidence that mercury could be causing these physiological/pathological changes. Some of the research was done on animals.

Suggestion: Vascular contributions to Alzheimer’s disease.

Response: The vascular papers described in the manuscript summarized many of the vascular changes occurring in AD, with evidence that mercury could be causing them.